

# Postglacial environmental changes in the northwestern Barents Sea caused by meltwater outbursts

Dhanushka Devendra[1]*, Natalia Szymańska[1], Magdalena Łącka[1], Małgorzata Szymczak-Żyła[1], Magdalena Krajewska[1], Maciej M. Telesiński[1], Marek Zajączkowski[1]

[1] Department of Paleoceanography, Institute of Oceanology, Polish Academy of Sciences, Sopot 81-712, Poland

*correspondence to:* Dhanushka Devendra (devendra@iopan.pl)

**Abstract.** The last deglaciation was marked by abrupt shifts between cold and warm states reflecting an integrated response to the gradually increasing summer insolation at northern
latitudes, changing ocean circulation, and the retreat of the Northern Hemisphere ice sheets. In this study, we present new multiproxy reconstructions of water mass properties and sea surface characteristics from a sediment core from the northwestern Barents Sea (Kveithola) representing the last 14,700 years. Our reconstruction documents four sediment-laden meltwater pulses between 14,700 and 8,200 cal years BP based on biomarkers, stable isotopes,
and sedimentological parameters. Deglacial processes primarily cause these meltwater pulses and are possibly supplemented with paleo-lake outbursts, paleo-tsunami currents, or a combination of at least one of these, are characterized by sudden drops in sea surface temperatures, increased sea ice formation, increased terrigenous supply, and a limited influence of Atlantic Water in the northwestern Barents Sea. The influence of the Storegga tsunami, which
occurred around the 8,200 cal years BP cooling event likely reached and redistributed the sediment in Kveithola. Strong coarsening of the northwestern Barents shelf was observed after 3,500 years, which might be related to a stronger Atlantic Water inflow from the west across the bank leading to winnowing.

**Keywords.** Meltwater; Atlantic Water; sea ice; northwest Barents Sea; Post-glacial



## 1.    Introduction

Meltwater pulses appear to have a distinct and dramatic impact on the climate and oceanography of the European Arctic (Nesje et al., 2004; Harrison et al., 2018; You et al., 2023). Studies reconstructing meltwater pulses and their repercussions gain particular significance in the context of modern climate change, as more meltwater events are anticipated (Harrison et al., 2018; Tuckett et al., 2021). Numerous recent modeling studies have attempted to anticipate the

consequences of meltwater discharge from the Antarctic or Greenland ice sheets (Ackermann et al., 2020; Alsos et al., 2022; Nobre et al., 2023; Li et al., 2023). Some predict a slowdown or shutdown of the Meridional Overturning Circulation which would result in severe climatic consequences for the Earth e.g., (Lohmann and Ditlevsen, 2021).

The primary source of deglacial meltwater pulses reaching the Nordic Seas are the

Fennoscandian and Svalbard-Barents Sea ice sheets. Uncertainties persist, however, regarding the timelines and intensity of meltwater events occurring through the deglaciation and early Holocene (Harrison et al., 2018; Lucchi et al., 2015; Paus et al., 2019). A study conducted at the depositional mouth fans west of Spitsbergenbanken has shown evidence of two meltwater pulses: MWP-1a0, approximately 19,000 years BP, and MWP-1a, approximately 14,500 years

BP (Lucchi et al., 2015). However, there remains a gap in our understanding of meltwater pulses during the Early and mid-Holocene periods.

Situated on the margin of the Barents Sea continental shelf, the Kveithola Trough was shaped by glacial processes (Fig. 1). The Kveithola Trough experiences intense water mass exchange. Its location and latitudinal orientation cause it to host AW flowing into the Barents Sea (Rumohr

et al., 2001). The trough also acts as a conduit for cold, saline bottom waters laden with sediment, which descend from Spitsbergenbanken to the continental slope (Fohrmann et al., 1998) (Fig. 1c). How these water masses interact varies based on the strength of AW advection, sea-ice distribution, and local oceanographic conditions.

Previous studies at the Kveithola Trough have focused on seismic stratigraphy e.g., (Rüther et

al., 2012; Rebesco et al., 2016; Bjarnadóttir et al., 2012). Paleoceanographic investigations exists from the eastern, upper section of the trough, covering the last ~12,000 years e.g., (Belt et al., 2015; Berben et al., 2014; Groot et al., 2014). Here we present a sedimentary record spanning the past 14,700 years from the lower, western, portion of the Kveithola Trough. The location, situated in the west of the depositional fan allowed for the acquisition of a record

preserved in a steady environment (Fig. 1). At the same time the western part of the trough is deep enough (>350m) to be unaffected by seabed erosion by icebergs (Lucchi et al., 2015). We



have implemented an array of proxies ranging from biomarkers to micropaleontological assemblages to best reconstruct the paleoenvironment and discern the presence and impact of meltwater pulses at Northwestern (NW) Barents Sea.

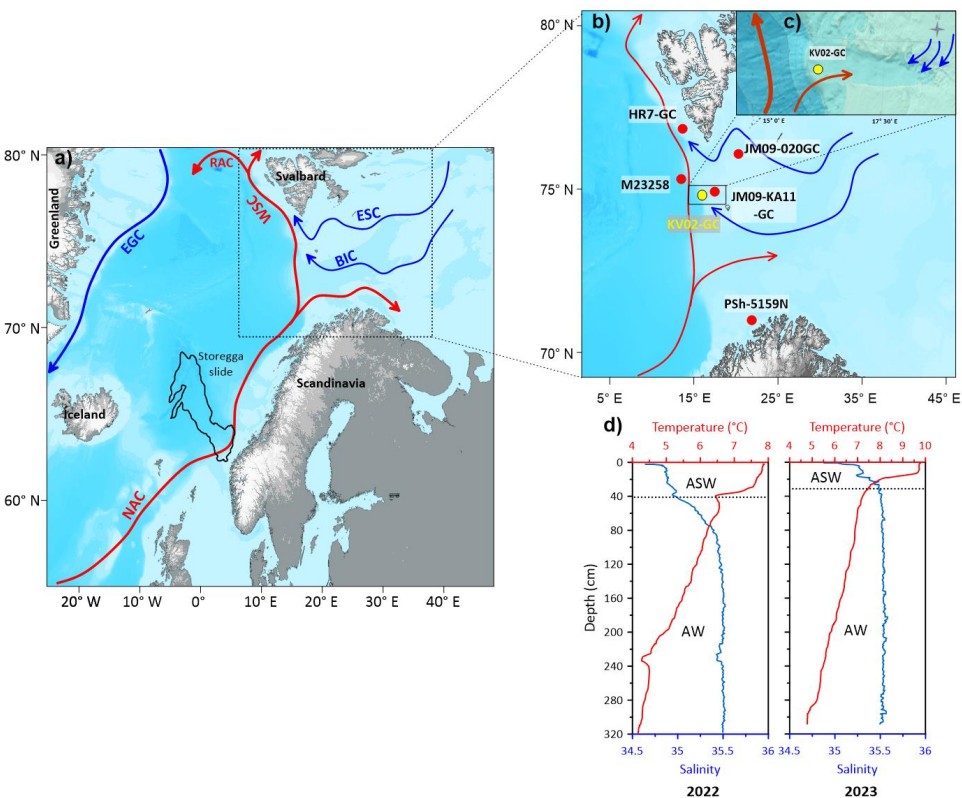


**Figure 1: (a) Present-day surface ocean circulation in the Nordic Seas. The black polygon shows the extent of the Storegga slide** (Bondevik et al., 2005)**. (b) Locations of marine sediment cores mentioned in the text: OCE2022-KV2-GC (this study), HR7-GC** (Devendra et al., 2023)**, JM09-020-GC** (Łącka et al., 2019)**, M23258** (Sarnthein et al., 2003)**, JM09-KA11-GC** (Berben et al., 2014) **and PSh-5159N** (Risebrobakken et
al., 2010)**. (c) Bathymetry map of the Kveithola Trough. The red arrows represent the flow of warm Atlantic Water, i.e., North Atlantic Current (NAC), West Spitsbergen Current (WSC), and Return Atlantic Current (RAC). The blue arrows represent the East Greenland Current (EGC), East Spitsbergen Current (ESC), and the Bear Island Current (BIC). (d) Salinity, temperature, and depth profiles were measured in August 2022 and August 2023 at the core site of OCE2022-KV02-GC (temperature in red and salinity in blue).**
**Abbreviations: ASW – Arctic Surface Water; AW – Atlantic Water.**

## 2. Regional setting

The western Barents Sea and west Spitsbergen continental margins direct the northward propagating Norwegian Atlantic Current (NwAC). The NwAC moves from the south, carrying





warm and salty Atlantic Water (T > 3 °C, S > 35.0) into the Nordic Seas (Smedsrud et al., 2021;
Hopkins, 1991). It then spreads into the Barents Sea through North Cape Current (NCaC), and
into the Arctic Ocean via the West Spitsbergen Current (Fig. 1). Relatively cold and less saline
Arctic Water (T < 3 °C, S < 35.0; Hopkins (1991)) coming from the Arctic Ocean is brought
into the Barents Sea and is carried southward by the East Spitsbergen Current (ESC) and Bear
Island Current (BIC) and then flows northward along the western coast of the Svalbard
archipelago (Loeng, 1991; Loeng et al., 1997). The water mass carried by ESC and BIC has
relatively low salinity and very low temperatures and is often covered by/loaded with seasonal
sea ice (Loeng et al., 1997).

Kveithola is a small trough located at the western Barents Sea margin, northwest of Bear Island
(Fig. 1). This trough is characterized by water depths ranging from 200 to 400m and is
surrounded by Spitsbergenbanken to the north, south, and east (Fig. 1). To the west, Kveithola
extends westward and terminates near the continental margin. Two oceanographic processes
are currently active in Kveithola. Firstly, Kveithola acts as a pathway for cold and saline bottom
waters that are sediment-laden and cascade down from Spitsbergenbanken to the continental
slope (Fohrmann et al., 1998). Secondly, the inflow of warm saline Atlantic waters is known to
affect Kveithola, as reported by (Rumohr et al., 2001). The bottom currents activities are likely
to vary based on the strength of Atlantic water advection, sea-ice distribution, and local
oceanographic conditions.

## 3.    Material and methods

### 3.1.    Core collection and CTD

Gravity core OCE2022-KV02-GC (hereafter KV02) was retrieved from the western Barents
Sea (74°50.296N, 16°1.3403E, 374 m water depth) during the AREX expedition with R/V
*Oceania* in summer 2022 (Fig. 1). The water column temperature and salinity at the coring site
were measured in situ by a mini-CTD (conductivity–temperature–depth) profiler at intervals of
1 s. After the sediment core opening, the visual description was performed and color
information was obtained based on the Munsell Soil Color Chart. Sediment is mostly light-
colored (olive-gray to grayish green) homogenous fine-grained mud and represents continuous
sedimentation with no sign of redeposition. The 1.42 m long core was sliced at 1 cm intervals,
freeze-dried, and wet sieved through 100 and 500 µm mesh.





**Table 1.** AMS $^{14}$C measurements and calibrated ages applying the Marine20 calibration curve. Median ages (in bold) were used for the age-depth model (Fig. 2). Calibrated ages are reported in thousand years before AD 1950 (kyr BP).

| Lab ID | Depth (cm) | Dated material | $^{14}$C age | Calibrated age (year BP, 2σ) | | |
|---|---|---|---|---|---|---|
| | | | | Min age | Max age | Median age |
| 11617.1.1 | 3 | Mixed benthic foraminifera | 2832 ± 81 | 2396 | 2658 | **2519** |
| 11618.1.1 | 13 | Mixed benthic foraminifera | 3795 ± 89 | 3544 | 3827 | **3684** |
| 11348.1.1 | 21 | Mixed benthic foraminifera | 6912 ± 78 | 7221 | 7422 | **7317** |
| 11619.1.1 | 39 | Mixed benthic foraminifera | 7698 ± 123 | 7926 | 8219 | **8077** |
| 11349.1.1 | 53 | Mixed benthic foraminifera | 23722* ± 278 | | Excluded | |
| 11620.1.1 | 61 | Mixed benthic foraminifera | 8199 ± 141 | 8436 | 8838 | **8647** |
| 11350.1.1 | 81 | Mixed benthic foraminifera | 9577 ± 107 | 10235 | 10552 | **10407** |
| 11621.1.1 | 111 | Mixed benthic foraminifera | 14287** ± 236 | | Excluded | |
| 11350.1.2 | 130 | Mixed benthic foraminifera | 12627 ± 143 | 13990 | 14530 | **14265** |
| 11622.1.1 | 137 | *E. clavatum* | 12811 ± 197 | 14208 | 14883 | **14548** |

* The radiocarbon age from 53 cm was excluded from the age-depth model (described in section 5.2). ** The radiocarbon age from 111 cm was excluded from the age-depth model (described in section 3.2)

### 3.2. AMS-$^{14}$C dates and chronology

The age model for core KV02 is based on eight radiocarbon dates obtained from specimens of

the seven mixed benthic foraminifera and one *Elphidium clavatum* sample dated at the MICADAS facility at the Alfred-Wegener Institute in Bremerhaven, Germany (Fig. 2, Table 1). The resulting raw radiocarbon dates were calibrated to calendar ages in the CALIB $^{14}$C software (8.2.0; Stuiver et al. (2022)) using the Marine20 dataset (Heaton et al., 2020), and applying a regional radiocarbon reservoir offset (ΔR) of -92 ± 34 years. This ΔR value of −92

± 34 years corresponds to 67 ± 34 years relative to the Marine04 calibration curve based on data from near Bear Island (Mangerud and Gulliksen, 1975). We used http://calib.org/marine/ (last accessed 2024/04/08) to obtain the ΔR value relative to the Marine20 calibration curve. The age obtained in KV02 at 111 cm is older than the age at 130 cm (Table 1). This age reversal was possibly caused by sediment reworking, and the older date was therefore excluded from

the age model.



### 3.3. Foraminiferal analyses

Seventy-six sediment samples from core KV02 were used for foraminiferal analyses. Where possible, a minimum of 300 benthic foraminiferal specimens from the >100 µm fraction of each sample were picked under the light microscope and mounted on faunal slides. However, samples analyzed between 93 and 101 cm had only a few specimens present (less than 15 individuals per sample) and were excluded from the statistical analysis (Fig. 3). The abundance of planktic foraminifera was very low throughout the core (less than 15 individuals per sample), except in the top 45 cm (Fig. 4). Therefore, we excluded these samples with fewer than 15 individuals from the calculation. When necessary, residues were split using a dry microsplitter, and the total number of foraminifera was calculated. The taxonomic classification was performed mostly at the species level, using the generic classification of Loeblich Jr and Tappan (2015).

The total benthic and planktic foraminiferal fluxes were calculated using foraminiferal number (FN) [N g$^{-1}$], sedimentation accumulation rate (SAR) [cm yr$^{-1}$], and dry bulk density (DBD) [g cm$^{-3}$]. The SAR was calculated by using the calibrated ages. The foraminiferal fluxes were calculated by using the following equation (1):

$$\text{Flux (individuals} \times \text{cm}^{-2} \times \text{kyr}^{-1}) = \text{FN} \times \text{DBD} \times \text{SAR}, \tag{1}$$

### 3.4. Stable isotopes and IRD

For stable carbon and oxygen isotope analysis, approximately 20 specimens of the planktic foraminiferal species *Neogloboquadrina pachyderma* and 25 specimens of the benthic species *E. clavatum* were picked from the 100–500 µm size fraction. All samples were cleaned in methanol, and measurements were performed using a Thermo Finnigan MAT 252 mass spectrometer with a Kiel III automatic carbonate preparation device at the Light Stable Isotope Mass Spec Laboratory, Department of Geological Sciences, University of Florida, USA. Measurement precision was better than ±0.04‰ and ±0.02‰ for oxygen and carbon isotopes, respectively.

For ice-rafted debris (IRD) content of grains larger than 500 µm were counted in 63 samples. The number of IRD/g dry sediment was calculated. The flux was calculated, using the concentration of IRD [N g$^{-1}$], SAR, and DBD.



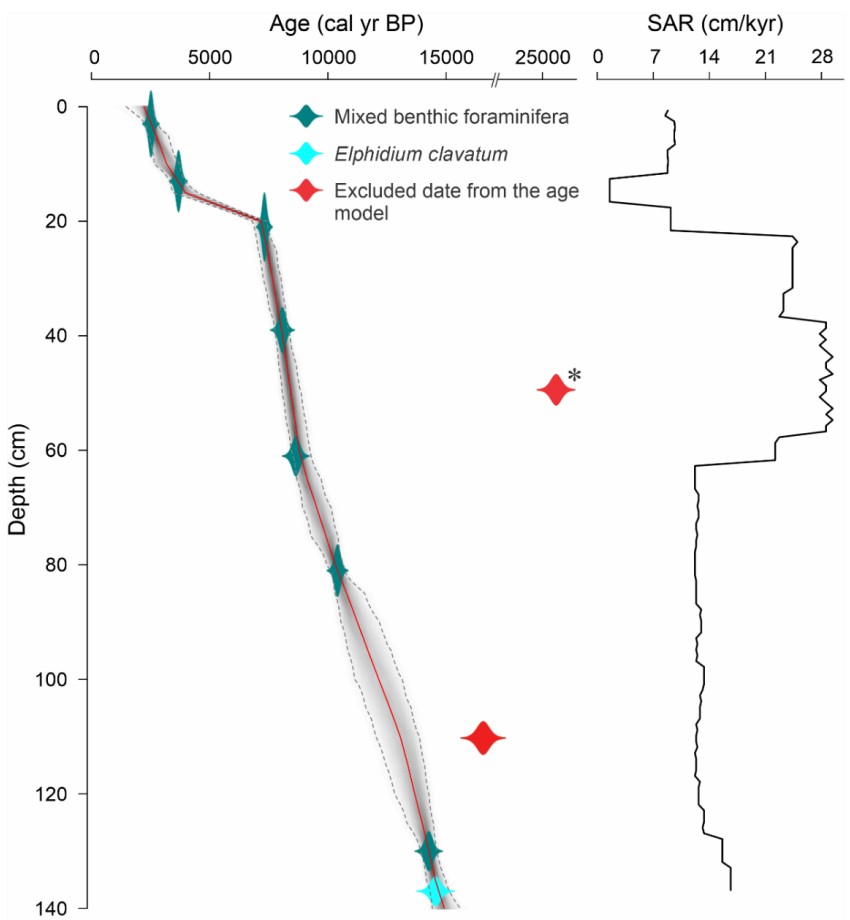

**Figure 2: Age-depth model of core OCE2022-KV02-GC. age-depth model and sedimentation accumulation**
165    **rates (SAR). (The asterisk next to the middle dated layer is described in section 6).**

### 3.5. Organic bulk sediment parameters and biomarker analysis

Total carbon (TC) and total organic carbon (TOC) were measured on homogenized bulk
sediment samples using the combustion technique with chromatographic detection, performed
170    with a Thermo Flash 2000 elemental analyzer. The TOC measurements were performed after
removing residual carbonate by adding hydrochloric acid. The carbonate content ($CaCO_3$) was
calculated according to TOC and TC values, using the following equation (2).

$$CaCO_3 = (TC-TOC) \times 100\backslash 12 \qquad\qquad (2)$$

Lipid biomarkers were analyzed at the Department of Paleoceanography, Institute of
175    Oceanology, Poland, using the procedures described in detail by Krajewska et al. (2023) and
Szymczak-Żyła and Lubecki (2022), with some modifications. Freeze-dried samples (~ 4 g),



spiked with surrogate standards (9-octylheptadecene, 7-hexylnonadecane, 2-nonadecanone, and androstanol) were sonication-extracted with a dichloromethane : methanol (2:1 v/v) mixture (3 x 15 ml). After extraction, the raw extracts were concentrated using rotary evaporation and divided into two subsamples (A and B).

Subsample A was fractionated by solid-phase extraction (SPE) using 1% deactivated silica gel. The fraction containing highly branched isoprenoid ($IP_{25}$) was eluted with a hexane and dichloromethane (1:1 v/v) mixture, whereas the fraction containing alkenones was eluted with dichloromethane. $IP_{25}$ were analyzed using a gas chromatograph coupled to a quadrupole mass spectrometer detector (GCMS-QP2010 Ultra; Shimadzu) according to the procedure described in Krajewska et al. (2023). The concentrations of $IP_{25}$ were calculated based on response factors for $IP_{25}$ and the standards following the procedure described by Belt et al. (2012). Alkenones were analyzed using a gas chromatograph with a flame ionization detector (GC/FID; Nexis GC-2030, Shimadzu) (Krajewska et al., 2023).

To determine the total (free + esterified) polar steroids, subsample B was saponified with 5% KOH in methanol, and then liquid-liquid extraction was carried out in the KOH–methanol extract/chloroform/water system (1/3/2 v/v/v; three times). The combined chloroform fractions were concentrated by rotary evaporation, derivatized to produce the trimethylsilyl derivatives, and analyzed using a gas chromatograph coupled to a mass spectrometer detector (GCMS-QP2010 Ultra; Shimadzu) according to procedure described in Szymczak-Żyła and Lubecki (2022). The analytes were identified based on their retention times and mass spectra and quantified based on response factors derived from daily injections of standard mixtures. The following polar steroids were studied: brassicasterol, campesterol, sitosterol, and dinosterol.

The $PIP_{25}$ indices were calculated by combining $IP_{25}$ with different phytoplankton marker brassicasterol for sea ice reconstruction using equation (3) (Müller et al., 2011).

$$P_pIP_{25} = IP_{25}/ (IP_{25} + (p \times c)) \qquad (3)$$

where $p$ is the phytoplankton marker concentration (brassicasterol (B)), and c is a balance factor (mean $IP_{25}$ concentration/mean $p$ concentration).

The $U^{K'}_{37}$ index was used to estimate the sea surface temperature (SST). The $U^{K'}_{37}$ index was calculated by the following equation (4) (Bendle and Rosell-Melé, 2004).

$$U^{K'}_{37} = [C_{37:2}]/([C_{37:2}] + [C_{37:3}] + [C_{37:4}]) \qquad (4)$$

$U^{K'}_{37}$ values were converted to SSTs using global core-top calibration using equation (5) (Müller



et al., 1998).

$$U^{K'}_{37} = (0.033 \times SST) + 0.044 \tag{5}$$


### 3.6. X-ray fluorescence spectrometry

X-ray fluorescence (XRF) scanning of the sediment core was conducted at the Department of Paleoceanography, Institute of Oceanology, Poland. An Olympus Vanta M series portable XRF analyzer was used to determine the concentration of the element in the samples. The instrument uses a Rh anode X-ray tube (8-50 kV) as the excitation source. The instrument was operated in

Geochem Mode with a scanning time of 45 seconds per beam. The instrument completed one whole scan in 90 seconds by scanning via two beams in sequence. For the interpretation, element ratios, rather than individual elements, are used to prevent closed-sum effects.

## 4. Results

### 4.1. Age model


The Bayesian accumulation age-depth modeling program, Bacon 2.4.3 (Blaauw and Christen, 2011) in R statistical software was used to create a suitable age model for sediment core KV02 (Fig. 2). Due to the gravity coring process ~2000 years of sediment have been lost. The calculated sediment accumulation rates (SARs) vary significantly throughout the core. High

SARs were observed in the lowermost part of the core, between 138-112 cm of sediment core (between 14.4 and 13.1 kyr BP; 10-15 cm kyr$^{-1}$), followed by a decrease between 112 and 63 cm (~13 to 8.7 kyr BP; below 10 cm kyr$^{-1}$). The highest SARs were observed between 62 and 23 cm (~8.7 to 8.1 kyr BP; reaching up to ~30 cm kyr$^{-1}$), followed by an abrupt decrease of ~1.5 cm kyr$^{-1}$ until 13 cm (up to ~3.4 kyr BP). Then, SARs increased from 13 cm to the surface,

reaching ~11 cm kyr$^{-1}$ (Fig. 2).

### 4.2. Sediment and geochemistry

Between 14.4 and 12.7 kyr BP, coarse content remained low but significantly increased by the end of this interval. From 11.5 to 10.7 kyr BP, the coarse content gradually rose and then stabilized until 9.5 kyr BP. A subsequent gradual increase led to a peak of around 6.8 kyr BP,

followed by a notable drop between 8.3 and 8 kyr BP. After 6.8 kyr BP, the coarse content decreased and remained stable until a sudden increase at 3.5 kyr BP (Fig.4). Fe and Ti show fluctuating trends, peaking during the periods of 14.4 - 12.9, 11.7 - 10.9, 10.4 - 9.6, and 9 - 7.5 kyr BP, coinciding with high IRD content (Fig. 4). The TOC percentages remained relatively stable (~1.4 wt%) until ~7.5 kyr BP, with few distinct fluctuations around 11, 9, and 7.5 kyr





BP. After 7.5 kyr BP, the TOC percentage rose noticeably (from 1.15 to 1.8 wt%), and remained constant throughout the rest of the core (Fig. 5).

### 4.3. Foraminifera

A total of 38 benthic foraminiferal species were identified from the samples counted (33 calcareous and 5 agglutinated). The abundance of agglutinated species was very low; no single

species had a relative abundance greater than 1% in any analyzed sample. Selected benthic and planktic foraminiferal species abundances are presented in Fig. 3 and 4 respectively. The benthic foraminiferal concentration varies from a minimum of <1 (three samples at 93, 95, and 101 cm) up to a maximum of 300 specimens g$^{-1}$.

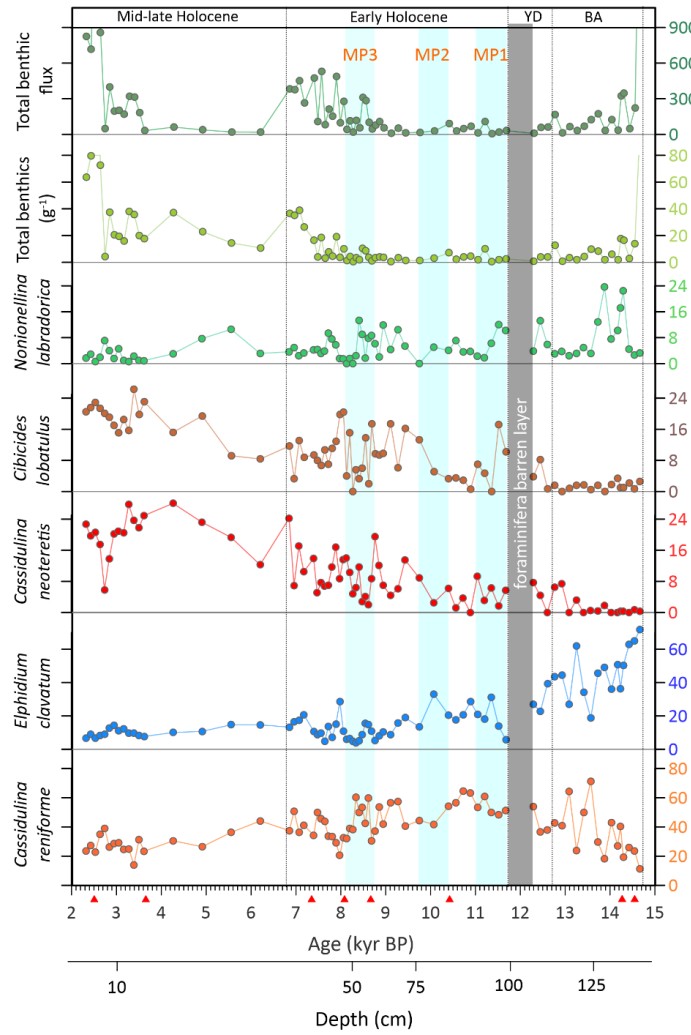




**Figure 3: Relative abundances of selected benthic foraminiferal species of core KV02.** *Cassidulina reniforme* (orange line); *Elphidium clavatum* (blue line); *Cassidulina neoteretis* (red line); *Cibicides lobatulus* (brown line); *Nonionellina labradorica* (light green line); total benthic foraminiferal concentration and flux (green lines). Blue shadings indicate meltwater pulses. Red triangles next to the bottom x-axis indicate radiocarbon
dates. Abbreviations: BA - Bølling-Allerød; YD - Younger Dryas.

The benthic fauna is dominated by four benthic foraminiferal species; *E. clavatum* (4-72%), *Cassidulina reniforme* (0-71%), *Cassidulina neoteretis* (0-30%), *Cibicides lobatulus* (0-26%) and *Nonionellina labradorica* (0-25%) (Fig. 3). Between 14.4 and ~13 kyr BP, *E. clavatum* and
*C. reniforme* dominate the fauna with average values of 47% and 35%, respectively. From 13 to 10.2 kyr BP, *E. clavatum* shows a gradual decreasing trend where the abundance of *C. reniforme* gradually increases. The abundance of *N. labradorica* shows an increasing trend between 12.6 and 11.4 kyr BP. Between 12.1 and 11.8 kyr BP, our record shows very low foraminiferal content. Two species, *C. neoteretis* and *C. lobatulus* show relatively low
abundance until 9.5 kyr BP, and from that point abundance increases gradually throughout the records. Similarly, the abundance of *I. norcrossi* gradually increased from 10 kyr BP and reached its maximum abundance between ~9 and 7.5 kyr BP, and then decreased until the end of the record. After 10 kyr BP, both *E. clavatum* and *C. reniforme* decrease gradually until the end of the record.


The polar species *Neogloboquadrina pachyderma* and subpolar *Turborotalita quinqueloba* and *Neogloboquadrina incompta* dominate the planktic foraminiferal assemblages throughout the entire records (Fig. 4). There is a decrease in the abundance of these polar species from ~8 to 7 kyr BP, during which the subpolar species exhibit relatively high abundance. After 7 kyr BP,
the polar species dominated the entire record (Fig. 4).

### 4.4.  Stable isotopes

The benthic foraminiferal $\delta^{18}$O profile shows highly variable values (+2.7 to +3.6‰) at the base of the core, between 14.7 and 12.5 kyr BP (Fig. 4). Between 12.5 and 8.5 kyr BP, the values are relatively stable with an increase (to positive) around 12.3 and 9 kyr BP. From 9 kyr BP, the
$\delta^{18}$O profile shows highly variable values (+3 to +3.9‰) until 6 kyr BP, with the most depleted value around 8.6 kyr BP (3‰). After 3.5 kyr BP, benthic $\delta^{18}$O becomes heavier before lightening again around 2.2 kyr BP. Benthic $\delta^{13}$C values remain generally stable until 11 kyr BP, then become lighter until 8.8 kyr BP (Fig. 5). The $\delta^{13}$C values are highly variable between 8.8 and 7 kyr BP, with the most depleted value recorded at 8.4 kyr BP. Then, values become



more stable before becoming further lighter after 3.5 kyr BP. Planktic $\delta^{18}O$ values vary between

1.90 and 3.02‰.

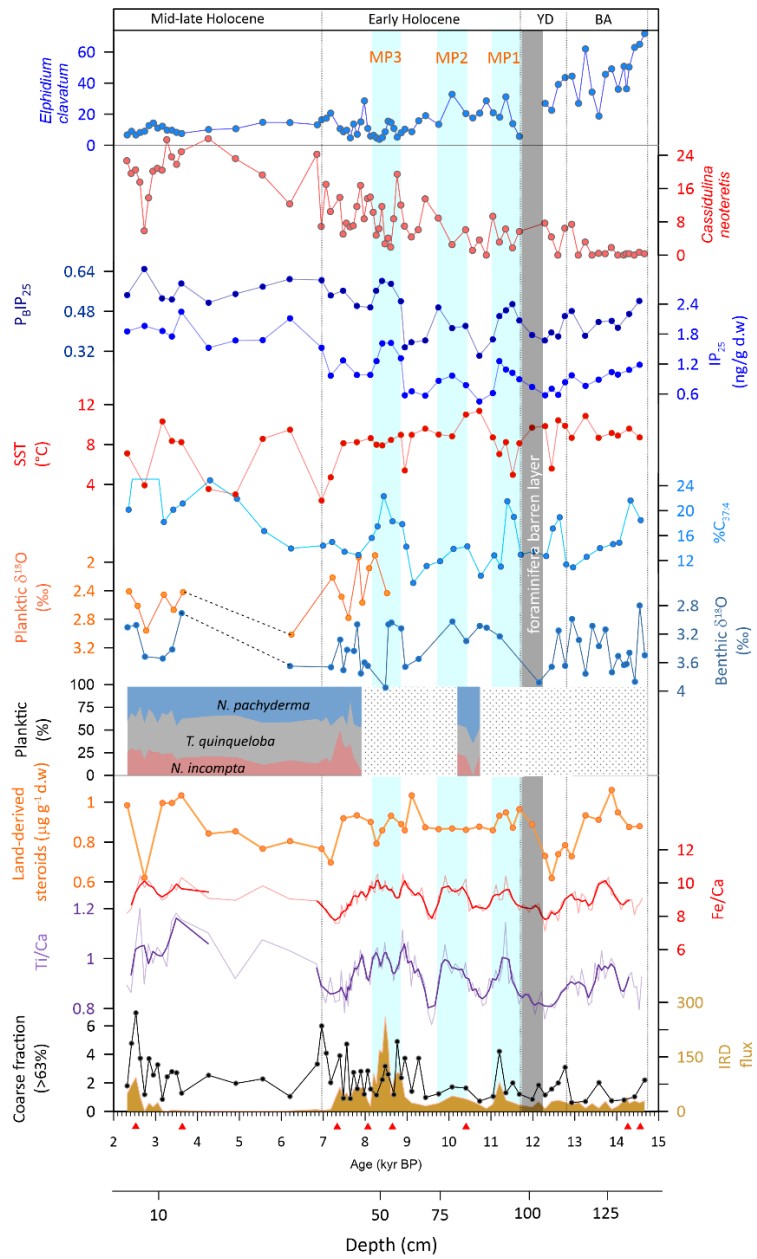

**Figure 4: Combined records of bulk parameters, foraminiferal, and stable oxygen isotope from core KV02.
From bottom to top: IRD content, % of coarse fraction, X-ray fluorescence (XRF) data (thin line = raw**
**data, thick line = three-point running average), land-derived steroids (ΣS-land = campesterol + stigmasterol
+ sitosterol + sitostanol), percentage of planktic foraminifera, benthic and planktic foraminiferal $\delta^{18}O$,**





percentage of tetra-unsaturated alkenone (%$C_{37:4}$), alkenone derived SST, concentration of IP$_{25}$, calculated P$_B$IP$_{25}$ indices, percentage of *Cassidulina neoteretis*, and percentage of *Elphidium clavatum*. Blue shadings indicate meltwater pulses. Red triangles next to the bottom x-axis indicate radiocarbon dates.
Abbreviations: BA - Bølling-Allerød; YD - Younger Dryas.

## 5.    Discussion

### 5.1.    Paleoenvironmental evolution during the last 14,700 years

#### Bølling–Allerød (B-A, ~14.7–12.7 kyr BP) – Initiated with a supply of meltwater

During B-A, the low diversity benthic foraminifera assemblage dominated by *E. clavatum* and
*C. reniforme* related to glaciomarine conditions with low temperatures and decreased salinities
(Łącka and Zajączkowski, 2016; Hald and Korsun, 1997), possibly caused by an immense
supply of meltwater. The high percentage of tetra-unsaturated alkenones (%$C_{37:4}$), an indicator
of low temperature and/or low salinity conditions, also suggests meltwater input to our site at
the beginning of the interval (Fig. 4) (Bendle et al., 2005; Bard et al., 2000; You et al., 2023).
The absence of planktic foraminifera and highly variable, but overall lighter benthic $\delta^{18}$O also
suggest the environment was mostly affected by high meltwater discharge (Fig. 4). There are
two possible sources of meltwater to our site. One is from the meltwater plumes triggered by
the disintegration of nearby ice sheets [Scandinavian Ice Sheet (SIS) and Svalbard Barents Sea
Ice Sheet (SBSIS)] and characterized by fine-grain suspended matter (Leng et al., 2018). The
other is melting icebergs, which are characterized by high IRD content. However, the IRD
content and coarse clast are low (Fig. 4), therefore, the latter can be ignored and meltwater was
supplied to our site by proximal ice sheets. According to Hughes et al. (2016), around 15 kyr
BP, both SIS and SBSIS had marine terminates, suggesting that meltwater could have been
flowing to the NW Barents Sea from these ice sheets. Alternatively, seasonal sea ice melting
could have also contributed to an increased supply of meltwater at our study site. Relatively
low IRD deposition and low foraminiferal content indicate increased sedimentation by
suspension settling from meltwater plumes (Knies et al., 2007), and possibly the presence of
sea ice preventing iceberg drifting (Dowdeswell et al., 2000) reflected in high P$_B$IP$_{25}$ values
(Fig. 4). High Fe/Ca and Ti/Ca values and concentrations of land-derived steroids (ΣS-land)
(Fig. 4) indicate enhanced transport of post-glacial terrigenous materials to the NW Barents
Sea, likely transported via the meltwater plume (Lantzsch et al., 2017). Therefore, these results
demonstrate the importance of suspension-rich meltwater plumes controlling the sedimentation
at our site during the B-A interval. Similar turbid meltwater plume-induced sedimentation
during the B-A interval has been reported in western Svalbard margins (Rasmussen et al., 2007;



Jessen et al., 2010), and NW Barents Sea (Rüther et al., 2012; Lucchi et al., 2013; Melis et al., 2018; Lantzsch et al., 2017) and could be associated with the meltwater pulse 1a (MWP-1a). However, even in the environment affected by turbid meltwater, which could significantly reduce the primary production in the water column by increasing light attenuation, our results indicate some productivity reflected in the high value of phytoplankton biomarkers, relatively

high benthic foraminiferal fluxes, and, slightly high and stable TOC (Fig. 5). The supply of terrigenous material from the melting likely supplemented the nutrient levels in the upper surface water mass and boosted surface production (Knies and Stein, 1998).

Initially, a low abundance of *C. neoteretis*, a typical AW indicator (Cage et al., 2021; Jennings

et al., 2004; Knudsen et al., 2004) suggests a weak influence from AW current to the study site (Fig. 3). Massive meltwater inflow could have caused a deepening of the halocline (Rasmussen and Thomsen, 2004), hence extending the relatively fresh and cold water to a greater depth. This probably allows warm AW to move further west from our study site.

**Younger Dryas (YD, ~12.7–11.7 kyr BP) - Less productive harsh environment**

Our age model suggests that the transition into cold Younger Dryas conditions from the B-A happened around 12.7 kyr BP (Fig. 4). Overall high benthic $\delta^{18}$O at the initial stage of the YD (~12.7 - 12.3 kyr BP) may reflect more saline bottom water at our site (Fig. 4). This could be attributed to the subsurface AW inflow, as suggested by the presence of *C. neoteretis* (Fig. 4).

The high values of %$C_{37:4}$ suggests meltwater input to our site during this interval (Fig. 4). This is also in agreement with findings from studies at the western, southwestern, and northern margins of Svalbard (Bartels et al., 2017; Rasmussen et al., 2007; Ślubowska et al., 2005). Low $IP_{25}$ concentrations (<0.55 ng/g d.w) along with the significantly low phytoplankton biomarker records indicate the presence of extensive sea ice cover at our study site (Fig. 5). This aligns

with the finding from the neighboring core located ~30 kilometres to the east (Belt et al., 2015). The IRD content drop after the B-A interval (Fig. 4), and the notable drop of coarse clast content suggests that the primary route of icebergs from the Barents Sea was obstructed by the extended sea ice cover (Forwick and Vorren, 2009).

Shortly after the 12.3 kyr BP, several proxies indicated that the NW Barents Sea experienced a very harsh environmental condition that lasted until 11.8 kyr BP. The notable decline of $CaCO_3$ values, relatively low phytoplankton biomarkers, and absence of both planktic and benthic





foraminifera indicate the extreme condition and low primary production (Fig. 5). A comparable

condition, characterized by foraminifera absent layer dated to the early part of the YD, was also

observed in a sediment core (recovered from 253 m water depth; core: JM09-020GC; Fig. 1)

from Storfjordrenna by Lacka et al. (2020). They collectively attribute this phenomenon to the

anoxic condition at the bottom. Considering the age uncertainty inherent in the age models, a

similar bottom anoxic condition may exist in NW Barents Sea margins during the YD.

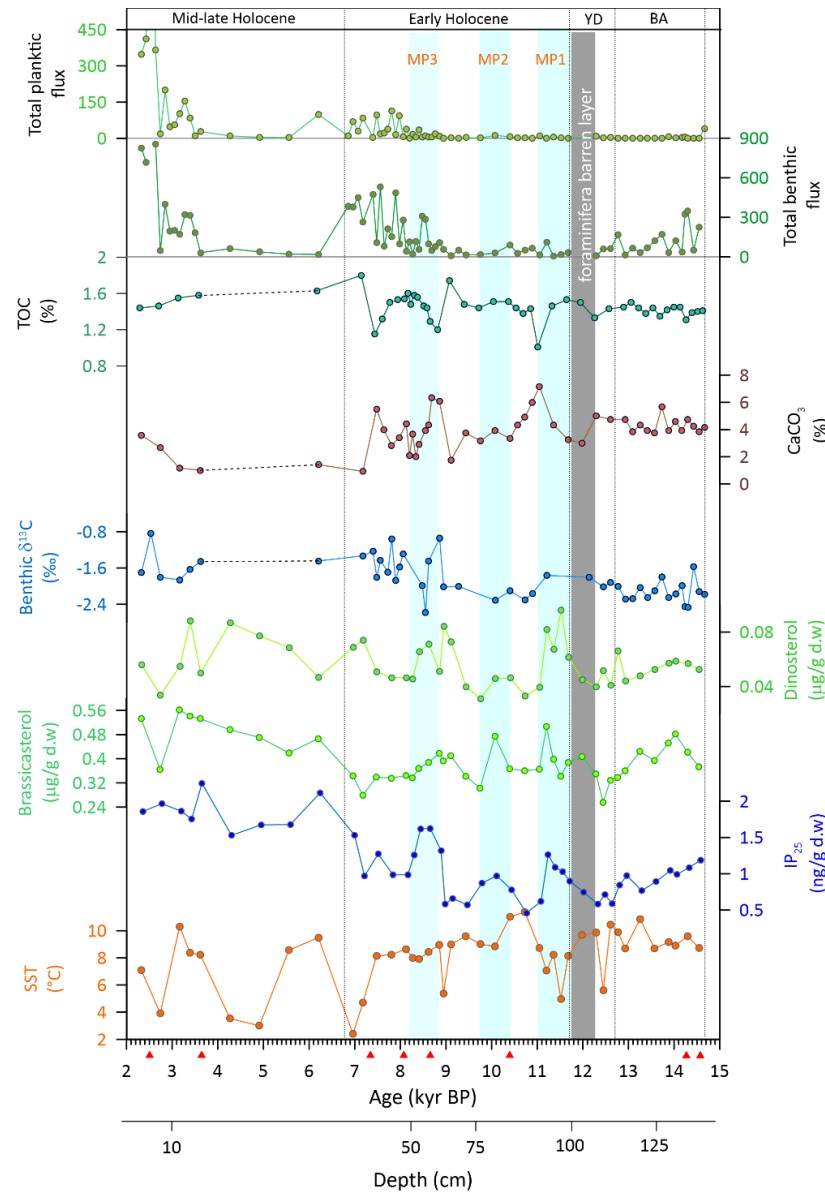




**Figure 5: Combined records of bulk parameters, stable carbon isotope, and foraminiferal from core KV02.
Bottom to top: alkenone-derived sea surface temperatures (SSTs), IP$_{25}$ concentration, phytoplankton
biomarkers (brassicasterol and dinosterol), benthic foraminiferal δ$^{13}$C, percentage of calcium carbonate,
percentage of total organic carbon, total benthic flux and total planktic flux. Blue shadings indicate
meltwater pulses. Red triangles next to the bottom x-axis indicate radiocarbon dates. Abbreviations: BA -
Bølling-Allerød; YD - Younger Dryas.**

### Early Holocene (∼11.7–6.8 kyr BP) – Dynamic environment characterized by short cold and warm spells.

Immediately after the YD, a prominent short-lived surface cooling was identified until ~11 kyr
BP. The surface was influenced by a cold meltwater plume, as indicated by the increased %C$_{37:4}$
(Fig. 4; for details see below section 6). This event is also marked by an increase in IP$_{25}$
concentrations, P$_B$IP$_{25}$ values, and decreased SSTs, altogether reflecting sea ice rafting (Fig. 4).
Relatively low abundance of *C. neoteretis* could suggest limited but some influence of AW to
our site, possibly as a subsurface water mass below the cold meltwater layer. The high
concentrations of phytoplankton biomarkers, increased CaCO$_3$, and TOC suggest high surface
productivity, possibly attributed to the freshly exposed terrigenous nutrients, transported via
meltwater. The sea surface condition just after YD is also recorded in the records from the
southwestern Barents Sea (core PSh-5159N; Risebrobakken et al. (2010)), northwestern
Barents Sea (core JM09-KA11-GC; Berben et al. (2014); Belt et al. (2015)), southern Svalbard
(core JM09-020GC; Łącka et al. (2015)) and western Svalbard shelf (core HR7-GC; Devendra
et al. (2023)), possibly associated with the cold event Preboreal Oscillation. During this period,
Arctic water conditions prevailed at most sites, and the Arctic Front was likely situated to the
west of the westernmost sites, specifically at the western Barents Sea and Svalbard margin.

Soon after this short cold period, planktic foraminifera appeared for the first time in the records
(Fig. 4). The planktic foraminifera fauna was dominated by subpolar *T. quinqueloba* (>40%),
occurred around 10.7 kyr BP and persisted until 10.3 kyr BP. A significant increase in subpolar
planktic species is also seen in the nearby core M23258 (Sarnthein et al., 2003; Risebrobakken
and Berben, 2018) and core JM09-KA11-GC (Berben et al., 2014) since ~10.5 kyr BP. This
suggests a short period of warm surface conditions with a low influx of meltwater to the NW
Barents Sea, also marked by higher SSTs, and abrupt drop in IP$_{25}$ and P$_B$IP$_{25}$ (Fig. 4). However,
low IP$_{25}$ still indicates the presence of some sea ice, probably seasonal and less extensive than
in the preceding interval. This condition was likely common to the region during this interval
(Rasmussen and Thomsen, 2015; Berben et al., 2014; Łącka et al., 2015; Devendra et al., 2023;





Belt et al., 2015).

The previous regional studies observed a distinct inflow of warm AW to the SW Svalbard and western Barents Sea margins around ~10 kyr BP (Rasmussen et al., 2007; Rasmussen et al., 2012; Ślubowska-Woldengen et al., 2008; Telesiński et al., 2018; Devendra et al., 2023; Risebrobakken et al., 2011). Similarly, an enhanced influx of AW to our study site was also suggested by increasing abundances of *C. neoteretis* initiated around 9.5 kyr BP (Fig. 3). The increasing abundance of *N. labradorica* observed from 9.3 kyr BP, suggest that the Arctic front was located close to our site (Fig. 3). Risebrobakken and Berben (2018) have also observed similar proximity of the Arctic front to the NW Barents Sea.

Between ~8.8 and 8.2 kyr BP, mild surface cooling associated with a distinct drop in AW influx and an abrupt increase in meltwater influx and sea ice cover reflected by the drop in *C. neoteretis* abundance and an increase in $\%C_{37:4}$ and, $IP_{25}$ and $P_BIP_{25}$ (Fig. 4; see also section 5.2). Subsequently, a slight warming of the surface ensued until 7.3 kyr BP, followed by another short period of pronounced surface cooling around 7 kyr BP, indicated by a drop in SSTs and an increase in $IP_{25}$ records (Fig. 4). A similar pronounced cooling trend around 7 kyr BP has been observed in several other records from the Nordic Seas (Knudsen et al., 2004; Rasmussen et al., 2007; Risebrobakken et al., 2011; Ebbesen et al., 2007; Werner et al., 2013; Werner et al., 2016; Hald et al., 2007).

**Mid-Holocene (~6.8–3.5 kyr BP) – Lag deposit associated with strong inflow of Atlantic Water**

The interpretation of the interval between 6.8 and 3.5 kyr BP is problematic due to the comparatively low temporal resolution caused by extremely low sedimentation rates (1.5 cm kyr$^{-1}$; Fig. 5) and, consequently, few data points (n=4). Lag deposit formation on Spitsbergenbanken due to strong winnowing induced by the enhanced influx of AW is likely to be the reason for this low sedimentation at our site corroborated by the previous studies (Vorren et al., 1984; Elverhøi and Henrich, 2002; Lantzsch et al., 2017). Following the trend from the latter part of the previous time interval, an enhanced influx of AW is indicated by the continuously increasing presence of *C. neoteretis* (Fig. 3). Planktic foraminifera fauna was dominated by subpolar *T. quinqueloba* (>40%) suggesting relatively warm surface conditions (Fig. 4). Support for similar warmer surface conditions after 7 kyr BP was also observed in SW Barents Sea (Risebrobakken and Berben, 2018) and the western Barents Sea (Berben et al., 2014). However, signs of sea ice expansion and fresher surface condition were suggested by



the increased IP$_{25}$ and P$_B$IP$_{25}$, and elevated %C$_{37:4}$ (Fig. 4). This suggests an inflow of AW flowing beneath this fresher surface water layer. A similar oceanographic condition with subsurface AW influx during the mid-Holocene has also been observed further north of the region (core: HR7-GC; Fig. 1) (Devendra et al., 2023). This may indicate that the Arctic front is located along the margin southwards from Svalbard, close to our study site, and is in agreement with Risebrobakken and Berben (2018).

**Late Holocene (after ~3.5 kyr BP) – Amelioration of the bottom environment by strong currents**

A drop in *N. pachyderma* abundance (~15%) and an increase in abundance of subpolar planktic species indicate that the surface water was characterized by relatively warm conditions (Fig. 4). Abrupt drop (to positive) in both benthic and planktic δ$^{18}$O records (Fig. 4) potentially due to the AW expansion over the entire water column. High P$_B$IP$_{25}$ (>0.5) values, along with the presence of IP$_{25}$, indicate a marginal sea ice zone or seasonal ice cover at our site (Fig. 4). These conditions with marginal sea ice significantly increased the annual productivity, which is reflected in high foraminiferal fluxes, CaCO$_3$, TOC, phytoplankton biomarkers, and lighter benthic δ$^{13}$C (Fig. 5).

This interval is also characterized by an increase in the content of coarse sediment at Kveithola (Fig. 4). Similar late Holocene coarser bottom conditions were identified from Spitsbergenbanken due to winnowing enhanced by bottom currents (Devendra et al., 2023; Andruleit et al., 1996). According to Sarnthein et al. (2003), pulsed inflows of AW to the western Barents Sea occurred after ~3 kyr BP, a timing that coincides with other observations in Fram Strait (Müller et al., 2012), western Svalbard (Devendra et al., 2023), eastern Svalbard (Pawłowska et al., 2020), and the SW Barents Sea (core PSh-5159N) (Risebrobakken et al., 2010). A significantly high abundance of *C. neoteretis* in our records also suggests an influx of AW to our site. Simultaneously, the high abundance of *C. lobatulus* that typically prefers high-energy environments with bottom currents (Wollenburg and Mackensen, 1998; Hald and Korsun, 1997; Mackensen et al., 1995) indicates that the AW current significantly influenced the environment. Therefore, we suggest that strengthening AW inflow and active bottom current-induced erosive conditions would have promoted the bottom coarsening condition within the region.

Alternatively, the coarsening could also be affected by both biogenic and terrigenous particle





influx (Lantzsch et al., 2017). The increase in biogenic carbonate content during this interval is reflected in high $CaCO_3$ content in our records (Fig. 5), which could be related to AW-induced

high surface water productivity, led to a decrease in terrigenous materials supply (Lantzsch et al., 2017) to our site. Our records show a decrease in terrigenous signals reflected in decreasing terrigenous biomarker and Fe/Ca (Fig. 4). Therefore, we suggest that the coarser sediment accumulation within the region would have been promoted by the combined effect of increased biogenic export from high surface annual productivity, and strengthening of AW inflow and

active bottom current induced erosive conditions.

### 5.2. Early Holocene meltwater outbursts to the western Barents Sea

We use the $\%C_{37:4}$ as an indicator of meltwater pulses at our study site. $\%C_{37:4}$ values are highly variable during the early Holocene, however, we observed three distinct increases in $\%C_{37:4}$

values (Fig. 6) that likely correspond to meltwater pulses (MP). We named these pulses MP1, MP2, and MP3, which occurred around 11.7–11 kyr BP, 10.4–9.7 kyr BP, and 8.8–8.2 kyr BP, respectively (Fig. 6). The absence of planktic foraminifera during these meltwater pulses may also support reduced salinity and turbid conditions in the surface water layer due to this turbid meltwater discharge (Fig. 4).


The occurrence of meltwater pulses was associated with preceding surface warming in the western Barents Sea (Fig. 6). The early Holocene meltwater pulses into the western Barents Sea were most likely controlled by different meltwater sources, such as 1) discharge from the disintegrating ice sheet from the northern part of Spitsbergenbanken to the inner Storfjorden, 2)

the meltwater outburst from paleo Ice Lakes, and 3) meltwater from the sea ice melting during the summer. These meltwater pulses were also marked by an abrupt increase in sea ice content (higher $IP_{25}$ values), a decrease in SSTs, and sluggish AW influx to the NW Barents Sea seen as the lower abundance of *C. neoteretis* and lighter benthic $\delta^{18}O$ (Fig. 6).

**MP1** is characterized by peaks in IRD content, land-derived steroids (ΣS-land), coarse clast content, and an increased Ti/Ca and Fe/Ca suggest an elevated supply of terrigenous materials, possibly transported by sediment-laden meltwater and icebergs and/or sea ice (Fig. 4). The observation of increased ice rafting in the study site coincides with higher IRD content in the NW Barents Sea (Lantzsch et al., 2017) and Storfjorden around 11.5 kyr BP (Rasmussen et al.,

2007). High drift sediment deposition from a disintegrating ice sheet from the northern part of





Spitsbergenbanken was found in Kveithola (Rüther et al., 2012), indicating that meltwater from an ice sheet from the northern part of Spitsbergenbanken flowed to Kveithola. Moreover, evidence of meltwater influence, as recorded in cores HR7-GC and JM09-020GC on the southern Svalbard shelf, supports the statement that the meltwater prominently originated from

the disintegrating ice sheet. The timing of this event compares well with the final retreat of the glaciers over the Svalbard and Barents Sea (Svendsen et al., 1996; Hughes et al., 2016). Alternatively, the outburst of Baltic Ice Lake to the North Sea (Fig.7a) around 11.5 kyr BP (Bjorck, 1995; Bodén et al., 1997; Nesje et al., 2004; Jakobsson et al., 2007), may have also contributed substantially to the development of MP1, considering the drainage timing. Hence,

we propose that the formation of MP1 is primarily attributed to the combined influence of meltwater discharge from ice sheets and drainage from Baltic Ice Lake. However, further research is required to understand and track the meltwater signals from the Baltic Lake outburst to the western Barents Sea.

**MP2,** which occurred between 10.4–9.7 kyr BP, is associated with lower volumes of meltwater influx or distribution to our study site compared to MP1. This is reflected in the relatively lower %$C_{37:4}$ during the MP2 than in the MP1 (Fig. 6), suggesting that at least one of the meltwater sources for the MP1 may have partially or completely stopped during the MP2. The meltwater originating from the northern part of Spitsbergenbanken likely partially or completely stopped,

supported by a significant reduction of meltwater influence on the southern Svalbard shelf (core HR07-GC; Devendra et al., 2023). After ~10.5 kyr BP, while the remnants of the last glacial ice sheet had retreated beyond the coastline of Norway, large parts of Fennoscandia remained covered by ice (Hughes et al., 2016). The orbital forced summer insolation was also high during this period (Laskar et al., 2004) (Fig. 6), which could promote the melting of the remaining ice

sheet and sea ice during the summer and meltwater release to the Barents Sea. Moreover, drainage from the Ancylus Lake (Fig. 7b) to the North Sea was also observed between 10.4 and 9.2 kyr BP (Bjorck, 1995; Nesje et al., 2004), and could have been transported to the western Barents Sea. Therefore, a substantial amount of turbid meltwater (reflected in higher Ti/Ca and Fe/Ca, and land-derived steroids; Figs. 4 and 6) could be transported to the western Barents Sea

due to the deglaciation process and possibly by the lake outburst flood.

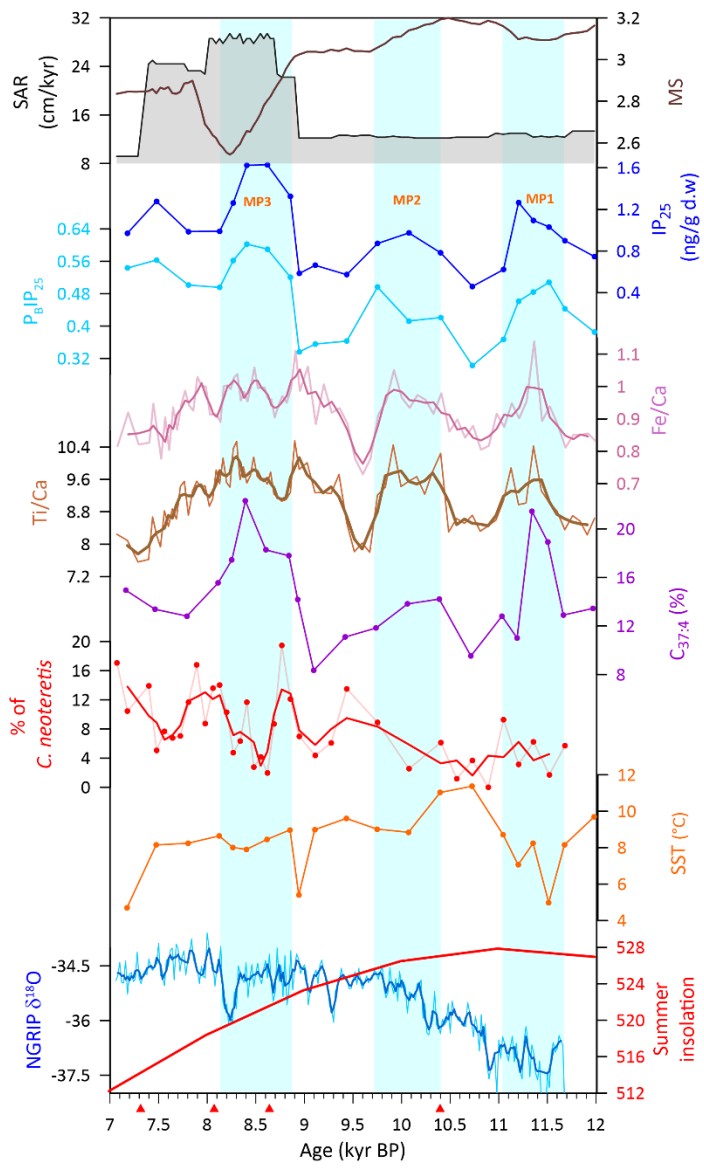

**Figure 6: Proxy records of early Holocene meltwater events. From bottom to top: δ¹⁸O of the NGRIP core (black line)** (Grootes and Stuiver, 1997) **and summer insolation at 77°N (red line)** (Laskar et al., 2004),

**alkenone-derived SST, percentage of *Cassidulina neoteretis*, percentage of tetra-unsaturated alkenone (%C₃₇:₄), Ti/Ca and Fe/Ca, calculated P_BIP₂₅ indices, concentration of IP₂₅, sediment accumulation rate (SAR) and magnetic susceptibility. Blue shadings indicate meltwater pulses (MP1, MP2, and MP3). Red triangles next to the x-axis indicate radiocarbon dates.**

**MP3** occurred between ~8.8 and 8.2 kyr BP, likely coinciding with an 8.2 kyr BP cold event (Barber et al., 1999; Johnsen et al., 2001; Von Grafenstein et al., 1998). Since all the marine





terminates glaciers around Svalbard were retreated to terrestrial margins and coasts remained ice-free through this interval e.g., (Farnsworth et al., 2020), the source of this meltwater is likely from the seasonal sea ice melting. The coincidence of low benthic carbon isotope values with this

meltwater pulse is consistent with developing a low-salinity surface layer that would increase stratification and inhibit ventilation (Fig. 5).

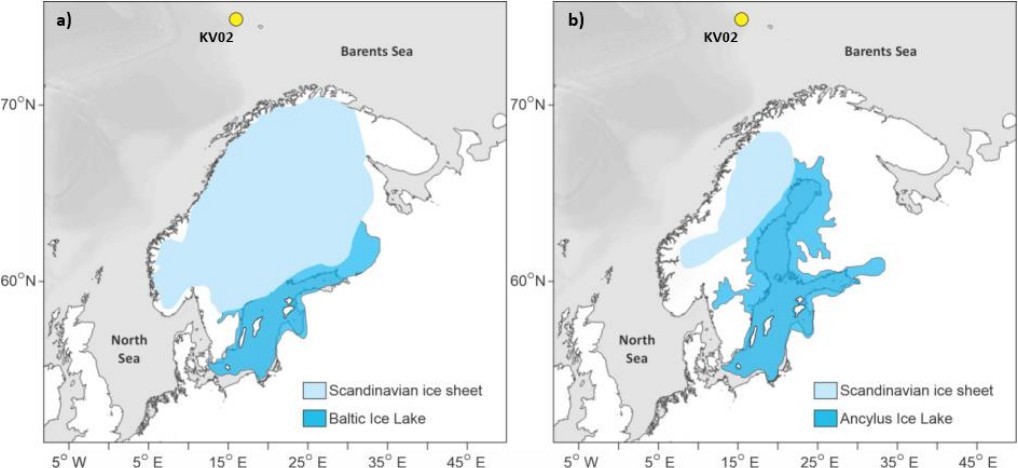

**Figure 7: The locations of the Scandinavian Ice Sheet (modified from Hughes et al. (2016)) extent and Paleo**
**Ice Lakes (modified from Harrison et al. (2018)). a) Baltic Ice Lake around 11.5 ka and b) Ancylus Ice Lake around 10 ka in present-day Fenno-Scandinavia. The yellow dots indicate the sediment core KV02.**

In addition to sea ice melting, turbid freshwater associated with high terrestrial materials could be transported from the Norwegian Sea via Storegga tsunami currents triggered by the Storegga

slide that occurred around 8.2 kyr BP in the Norwegian Sea (Fig. 1a). Within the Norwegian Sea, this turbid water, characterized by its freshness and abundance of terrestrial materials, may be further influenced by tsunami backwash (Bondevik et al., 2012). Rüther et al. (2012) speculate that the Storegga Tsunami currents may also have affected the NW Barents Sea. The computer simulation suggests that strong flow velocity generated by tsunami currents could

reach and disturb the sediment up to 74 °N (Bondevik et al., 2024). The abrupt decrease in magnetic susceptibility through this part of the record (supplementary Fig. 1) would suggest a sudden change in the sediment transport to the area. A high sediment accumulation (Fig. 6) during this interval, also a significant drop in sand content, and an increase in fine-grained sediment (supplementary Fig. 1) suggest that sediment suspension would play a vital role in

sedimentation at Kveithola. An age reversal (marked by an asterisk in Fig. 2) between the two




radiocarbon-dated layers (8.07 and 8.65 kyr BP), potentially due to sediment reworking and redeposit of the older foraminiferal shells to our site. This could be related to Storegga slide-induced current influence on the western Barents Sea reported by Rüther et al. (2012). Unfortunately, the exact timing of the influence of Storegga tsunami currents on the Kveithola
cannot be determined in our records due to the limited number of radiocarbon dates in this interval. Nevertheless, changes in the sedimentary environment during this interval (Fig. 6 and supplementary Fig. 1), consistent with the Storegga tsunami records in the region, lead us to conclude the NW Barents Sea could have also been affected by the tsunami currents, which brought meltwater to the NW Barents Sea.


### 6.    Conclusions

The multiproxy reconstruction using a sediment core from the northwest Barents Sea, representing the last 14,700 years, shows variations in water mass properties and sea surface features. We identified four sediment-laden meltwater pulses to the western Barents Sea
between 14,700 and 8,200 cal years BP. These meltwater events are characterized by sudden drops in sea surface temperatures, increased sea ice formation and terrigenous supply, and limited influence of Atlantic Water on the northwestern Barents Sea. The impact of the Storegga tsunami, which occurred around the 8,200 cal years BP cooling event is likely to have redistributed the sediment in Kveithola. Furthermore, the observed coarsening of the
northwestern Barents shelf after 3,500 cal years BP and a dynamic environment characterized by increased Atlantic Water influx demonstrate the complex interaction of processes shaping the region's environmental evolution. Overall, our findings advance our knowledge of the last deglaciation in the northwest Barents Sea and have implications for future studies into climate change and the reconstruction of the paleoenvironment in this region.


**Author contributions.** DD, NS, MMT, and MZ collected the sediment core. DD and MZ designed the research and experiments. DD constructed the age-depth model. DD, NS, MŁ, MS-Ż, and MK performed the formal analysis. DD wrote the original manuscript draft. All authors contributed to data interpretation, writing, and editing of the final manuscript.


**Competing interest.** The authors declare that they have no conflict of interest.



**Acknowledgments.** We would like to thank Agnieszka Kujawa for helping with the foraminiferal identifications. We would like to thank Bjørg Risebrobakken for her valuable suggestions to improve this manuscript. We also extend our gratitude to the R/V Oceania crew who helped during sediment core retrieval.

**Financial support.** The research was financially supported by the Norwegian Financial Mechanism for 2014-2021, project no 2019/34/H/ST10/00682. MŁ, MZ, and NS contributions were supported by the National Science Centre in Poland through project 2022/45/B/ST10/02033. Support for this work also came from the Johanna M. Resig Fellowship granted to DD by the Cushman Foundation.

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
