# Peer review of "Postglacial environmental changes in the northwestern Barents Sea caused by meltwater outbursts"

_Climate of the Past, 2024_

## Author Comment (AC1)

**Response to RC 1**

*All the responses are presented in italic blue color.*

*Thank you for your prompt review of our manuscript.*

The paper by Devendra et al. "Postglacial environmental changes in the northwestern Barents Sea caused by meltwater outbursts " attempts to reconstruct the paleoenvironment and the depositional regime from a single core from the trough Kveithola which is located at the western Barents Sea shelf and dates back to about 14.5 ka. To do so they employ quite a number of different proxy data. These include studies in micropalaeontology (benthic and planktonic foraminiferal assemblages and their O-isotopic composition), sedimentology (grain size fractions), geochemistry (XRF), and a suite of biomarker tools. They use all of these data to focus on 3 events, which they identify to be meltwater-triggered, and then, eventually, compare/relate the youngest of those to overregional changes far away.

While the paper is well written, its main focus, namely reconstructing meltwater outbursts during the first ~5ky of the Holocene from these proxies, is fundamentally flawed and therefore not acceptable for publication. The main question, apart from a solid age framework (see below), is to what extent the different proxies add up and can provide a consistently coherent story that also delivers some concrete information on the actual causes of their meltwater events? Instead the authors remain rather vague on this issue, naming all sorts of processes, from far-distance palaeolake drainage of the Baltic basin to tsunami-triggered currents by the Storegga slide to sediment-laden meltwater plumes....

*We find the reviewer's comments constructive and helpful, though we kindly disagree that the work is "fundamentally flawed". We are confident in the robustness of the approach and methodology we applied in this study. The interpretations are based on a comprehensive multiproxy analysis, and the conclusions drawn are well-supported by all the presented data. We think the evidence presented in the manuscript warrants our findings and assists in understanding meltwater outbursts during the discussed periods.*

*One of the main aims was to reconstruct past environmental changes in the NW Barents Sea with an emphasis on meltwater pulses (MPs) during the postglacial period. Reconstructing such events, especially from a single core, obviously poses challenges on its own. Nonetheless, the multidisciplinary nature of our study which includes micropaleontology, sedimentology, geochemistry, and biomarkers provides a firm ground for interpreting these past events.*

*Although each proxy may have its limitations, the strength of our study lies in integrating these different proxy data. By integrating the evidence of different proxies, we recovered a general picture of the paleoenvironmental events and depositional mechanisms in our study area. **This is discussed extensively in the original manuscript** ( Lines: 364-368, 477-484, and 516-520), where we integrated our results and compared them with the existing findings to increase the clarity and robustness of our conclusions.*

*We understand that the interpretation of the possible causes of meltwater pulses can be complicated. However, in this analysis, we have carefully thought about multiple potential processes. The identification and interpretation of these meltwater events are supported by a combination of different proxies (geochemical and sedimentological), which indicate*

*substantial inputs of freshwater and sediment during the periods in discussion. Lines: 301-306, 358-364, 392-394, 472-474, 503-505 etc..)*

*We have proposed that these meltwater pulses could be related to various processes; including regional glacial/sea ice melting, paleolake drainage, and even tsunami-triggered currents. Although it may sound broad, it reflects the complexity of interpreting paleoenvironmental signals in such a dynamic setting. Given the possibility that any of these processes may have been responsible for the observed changes, we aimed to present a range of possibilities rather than oversimplify the interpretation. We are convinced this is an advantage of the presented manuscript, not its disadvantage.*

*The question the reviewer posed, regarding whether the proxies tell a consistently coherent story. We understand the importance of ensuring that all proxies have to be coherent. In our study, we have carefully examined the data in each proxy to ensure that the data in the various proxies is consistent with one another.*

*For instance, the biomarker records, benthic foraminiferal assemblages, and their isotopic composition offer independent lines of evidence that indicate changes in water mass properties and meltwater input. The absence of planktic foraminifera during these meltwater intervals confirms our interpretation of sediment-laden meltwater influx. The geochemical data, including XRF measurements, are consistent with this occurrence of meltwater events as indicated by changing elemental compositions associated with significant terrigenous influxes around the meltwater events. All these results agree with the sedimentological data, which suggests a grain size distribution shift consistent with meltwater inputs and associated sediment-laden plumes. Since a major portion of our discussion covers this topic, we have not provided specific line numbers as references in this instance.*

Age model and associated problems: The 142cm long core has 10 radiocarbon dates, the lowermost 2 being of similar age. Unfortunately for the interpretation there are also 2 reversals, the lower one covers a huge age range (~4ky), and with 50cm amounts to more than 1/3 of the entire core length. And still, all proxy data from this section are being interpreted, including the YD interval and their MP1.

*We built the age-depth model here by using Bayesian statistics with the Bacon package in R (Blaauw and Christen, 2011). Bayesian statistics uses more sophisticated approaches, including constraining the accumulation rates and their variability, hence allowing the detection of "shift outliers" in the radiocarbon data. All this is to ensure that an age-depth model is robust and reliable.*

*We compared our multi-proxy records from the undated section (dotted area in Fig. 1; including YD and MP1) with the proxy records from nearby (~45 km to east) sediment core JM09-KA11-GC (using a new age model based on the Marine20 dataset). The trends in the multi-proxy records are consistent between the cores. This alignment provides support for our age model, especially in the section where direct dating is unavailable. We will include this comparison figure in the revised manuscript and provide a discussion on this as needed.*

*We believe the proxy data from this section provide valuable insights. We interpreted the YD interval and MP1 with careful consideration of the age model uncertainties. In particular, we cross-referenced our findings with existing regional records to ensure consistency in the broader context of paleoenvironmental changes. For instance, the timing of this MP1*

*compares well with the final retreat of the glaciers over the Svalbard and Barents Sea (Svendsen et al., 1996; Hughes et al., 2016), which could release meltwater to the Barents Sea (lines 482-483). Our interpretation is made with an understanding of the potential limitations and is framed within a range of possible scenarios.*

[Figure]

*Fig. 1. Comparison between proxies selected from this study and proxy records from sediment cores JM09-KA11-GC (Berben et al., 2014). Blue shadings indicate meltwater pulses.The dotted area indicates the section where direct dating is unavailable. Abbreviations: BA - Bølling-Allerød; YD - Younger Dryas.*

The second reversal at depth 53cm is interesting, because of its very old radiocarbon age which deviates from the dates below and above by about 15ky. It indicates that the bulk of forams measured were mostly very old, perhaps 30-40ka, and an admixture of just a few much younger ones caused the final age.

*Our data indicate a supply of older sediments from the upper part of Kveithola, which is confirmed by data from nearby core JM09-KA11-GC (Rüther et al., 2012; Groot et al., 2014; Berben et al., 2014).*

The crucial thing here is that depth 53 cm is in dead center in their MP3 (around their 8.2ka event), meaning whatever the cause, there was plenty of reworking and delivery of older sediments to the site, including all the fine TOC-rich sediments that would contain their biomarker proxies. The section of MP3 between the 2 acceptable dates is 20 cm long, telling us, together with the first reversal, that half of the 142cm long core (70cm) essentially remained undated. Moreover, the XRF-data (Fig. 4) across MP3 seems to imply that a larger sediment section was affected than indicated by the blueish bar.

*The dates used for constructing the age model around the reversal at 53 cm, specifically from 39 cm and 61 cm depths, firmly support our discussion and data interpretation. The meltwater signal, consistently indicated by all proxies, shows that the meltwater events (MP3) in the western Barents Sea occurred before the sediment reworking caused by the Storegga tsunami-induced currents. Therefore, the interpretation of this meltwater signal starting from 8.8 kyr BP is based on biomarkers and corroborated by other proxies, remains robust and well-founded.*

**Table 1.** AMS $^{14}$C dates and calibrated radiocarbon ages of JM09-KA11-GC. The calibration is based on the Marine13 calibration curve (Reimer et al., 2013) and a regional $\Delta R$ of $67 \pm 34$. The dates which are not used in the final depth–age model are indicated in italics.

| Lab ID | Core depth | Material | $^{14}$C yr BP (uncorrected) | $1\sigma$ | cal yr BP | $2-\sigma$ range | Reference |
|---|---|---|---|---|---|---|---|
| Tra-1063 | 4.5 cm | Mollusc dextral part of *Bathyarca glacialis* | 925 | 30 | 476 | 397–555 | Rüther et al. (2012) |
| *Tra-1064* | *4.5 cm* | *Mollusc dextral part of Bathyarca glacialis* | *900* | *35* | *445* | *354–535* | *Rüther et al. (2012)* |
| Tra-1065 | 16.0 cm | Mollusc sinistral part of *Bathyarca glacialis* | 1880 | 35 | 1377 | 1268–1485 | Rüther et al. (2012) |
| Beta-324049 | 27.5 cm | Benthic foraminifera *Islandiella norcrossi/helenae* | 4820 | 30 | 5027 | 4856–5197 | This study |
| *Tra-1066* | *33.0 cm* | *Mollusc dextral part of Astarte elliptica* | *1990* | *35* | *1469* | *1347–1590* | *Rüther et al. (2012)* |
| Beta-315192 | 40.0 cm | Benthic foraminifera *Islandiella norcrossi/helenae* | 5870 | 30 | 6211 | 6108–6313 | This study |
| Beta-315193 | 44.5 cm | Benthic foraminifera *Islandiella norcrossi/helenae* | 6890 | 40 | 7339 | 7241–7436 | This study |
| Tra-1067 | 55.0 cm | Mollusc sinistral part of *Astarte sulcata* | 7630 | 45 | 8038 | 7919–8154 | Rüther et al. (2012) |
| Beta-315194 | 80.5 cm | Benthic foraminifera *Islandiella norcrossi/helenae* | 9140 | 40 | 9790 | 9573–10 006 | This study |
| *Tra-1068* | *82.5 cm* | *Mollusc paired shell of Astarte elliptica* | *8140* | *50* | *8541* | *8387–8695* | *Rüther et al. (2012)* |
| *Tra-1069* | *82.5 cm* | *Mollusc sinistral part of Nuculana minuta* | *8315* | *50* | *8780* | *8595–8965* | *Rüther et al. (2012)* |
| *Beta-315195* | *111.0 cm* | *Benthic foraminifera Elphidium excavatum* | *10 900* | *50* | *12 309* | *12 072–12 546* | *This study* |
| Tra-1070 | 134.5 cm | Mollusc paired shell of *Yoldiella intermedia* | 10 705 | 55 | 11 993 | 11 668–12 318 | Rüther et al. (2012) |

*Radiocarbon dating from the core JM09-KA11-GC (Berben et al., 2014) also observed age reversal between ~8.1 and 8.6 kyr BP (Table. 1; from Berben et al., (2014)) which is compatible with our findings. Moreover, Rüther et al., (2012) observed the occurrence of mud chips in the sands during this interval, indicating a relatively strong impact, and suggested that the submarine Storegga slide and subsequent tsunami might be the cause. We have discussed this with our findings in the manuscript (Lines: 533-534). Given these similarities, we are confident that our age model is robust and well-supported by independent studies in the region. However, we plan to conduct additional dating between 7.9 and 8.4 kyr BP (between 39 and 61 cm) to better determine the thickness of the redeposited layer associated with the tsunami currents.*

*In conclusion, We do not agree with the reviewer's assessment regarding our age model. We used the best possible age model for this study. We have made a careful effort to develop an accurate chronology for the core based on radiocarbon dating by the Bayesian age model. This approach has allowed us to integrate multiple radiocarbon dates, account for potential outliers, and reduce the overall uncertainties in the age model. The Bayesian model has reduced error and created a more robust age-depth relationship.*

Speaking of the biomarkers, in the polar ocean elevated tetra-unsaturated alkenones (Uk37:4) are traditionally indicative for freshened waters and admixture of sediments which both could be derived from ice rafting. However, what is very critical using these samples with high 37:4 to be cautious when calculating SSTs from the 37:2 and 37:3. Indeed, the SSTs do not provide any consistency for huge temperature drops as claimed by the authors.

*The explanation given by the reviewer on $C_{37:4}$ coincides precisely with what we have presented and interpreted (with caution) in our manuscript (lines 288-289, 455-457). The comparison of the SST record with the nearby core shows very similar temperature variations, even though the two studies used different methods to derive SSTs. This consistency, particularly during the MPs (Fig. 1) strongly supports our data and interpretations. We will discuss this in our revised manuscript.*

*It is necessary to emphasize that our SST reconstructions are based on a careful analysis of the full suite of alkenones, including $C_{37:2}$, $C_{37:3}$, and $C_{37:4}$, with an understanding of the complex interactions in polar environments. Moreover, for the SST calculations, we used different $U^{K}_{37}$ indexes and different calibration equations. Finally, we used the $U^{K'}_{37}$ index, which is calculated as $U^{K'}_{37} = [C_{37:2}]/([C_{37:2}] + [C_{37:3}] + [C_{37:4}]$ (Bendle and Rosell-Melé, 2004). To obtain temperature values, we applied the equation developed by Müller et al. (1998) which provided the most realistic temperature estimates for our dataset.*

*We did not claim a **huge** temperature drop during the meltwater pulses or elsewhere in the discussion. Our SST reconstructions do not display significant cooling during the MPs, most importantly during the MP2 and MP3. However, these two pulses occurred in the warmer Holocene conditions and thus the observed drop of SSTs still upholds the story of meltwater discharge. Hence, the variability that was displayed in our SST records is in line with the impacts that could have been subjected to meltwater pulses, ice rafting, and other deglacial activities. Moreover, the SSTs we report are corroborated by other proxies (IP$_{25}$, Atlantic Water indicator proxies… etc.), ensuring that our interpretations are not based on alkenone data alone but on a holistic view of the paleoenvironmental conditions. These issues are discussed extensively in our manuscript, lines: 359-362, 375-387, 395-400, 467-470.*

Foraminifers: The authors claim that the entire core is dominated by planktonics...but my pdf only shows their occurrence after 8ka – incomplete because something wrong with Fig.4? The Barents Sea is known to be prone to massive calcite dissolution and I wonder of the authors noted some drastic changes throughout the core.

*We respectfully disagree with the reviewer's comment. We did not claim that planktic foraminifera dominate the entire core! On the contrary, we stated that planktic foraminifera are present in very low abundances throughout the record. The planktic foraminifera assemblages are dominated by three species (Figure 4), the polar species Neogloboquadrina pachyderma, and the subpolar species Turborotalita quinqueloba and Neogloboquadrina incompta as described in lines 130–133 and 258–260 of our manuscript. These species are spread only within the upper ~45 cm of the core. The individual count was quite low before the 45 cm (before 7.9 kyr BP, with about 10-15 individuals per sample) except for the layers between 77 and 85 cm (between 10.2-10.8 kyr BP). Therefore we excluded these samples with fewer than 15 individuals from the calculation. This exclusion has been explained in the manuscript (Lines 130-133).*

*The foraminiferal-barren layers were detected in the core (emphasized in all figures), which means neither planktic nor benthic foraminifera were discovered here. This barren layer was mentioned in the text (lines: 251, 340-345), and we discussed its implications in the context of environmental conditions such as an anoxic bottom environment, which is indeed a known phenomenon in the Barents Sea region e.g., (Lacka et al., 2020). The occurrence of this barren layer further supports our interpretation of the challenges associated with foraminiferal preservation in this area.*

*Regarding the visibility of planktonic foraminifera in Figure 4, we would like to clarify that planktonic foraminifera is absent before ~7.9 kyr BP (below ~45 cm; except layers between). Therefore, we keep a dotted background for the area of planktic foraminifera absent.*

The oxygen stable isotopes measured on the foraminifers appear very inconclusive with no proper trends over the entire core; the planktonic are only based on 15 samples and measure after MP3. The benthics vary over a 1 permil range and show no relation to any of the 3 MPs in terms of O-depletions. This is somewhat surprising considering that the authors also talk about sediment-laden plumes associated with these meltwater events.

*The analyses of stable isotopes in the planktonic foraminifera were conducted only in the layers where these foraminifera were present. Since we do not use the planktic $\delta^{18}O$ results for our interpretation (except line 426, which will be revised in the revised manuscript), we decided to remove the planktic $\delta^{18}O$ curve from Fig. 4. This change does not impact our interpretations of the MPs and the conclusions we made.*

*The benthic oxygen isotope values vary over a range of approximately 1‰, and in our opinion, this variability is expected in the context of the complex sedimentary and oceanographic processes in the region. In our opinion, it is seen in Fig. 4, especially in MP3 and MP2. The weak relationship between benthic isotopic depletion and all the meltwater pulses (MPs) on the surface reflects the interplay of multiple factors, including varying sources of meltwater, changes in water mass properties, and the dynamic sedimentary environment. Isotopic signals can be influenced by multiple factors masking the direct impact of meltwater events, particularly in records with lower temporal resolution.*

Sedimentology: I am puzzled about the use of grain sizes. In Fig. 4 the Coarse Fraction (>63µm), which to my mind is representative of the sand size fraction, does not exceed 5-7%. In Suppl-Fig, there is a completely different "sand" curve shown…

*Yes, we agree with the reviewer. The discrepancy between the coarse fraction (>63µm) in Figure 4 and the "sand" curve in the supplementary figure is due to incorrect scales used. We apologize for this oversight and acknowledge that it has led to some confusion. The correct representation of the sand-sized fraction will be included in the revised Figures. However, we would like to point out that this correction does in no way alter the interpretations or conclusions of this study.*

Concerning the figures in general there were no plots that would give the reader a better insight and understanding of the main proxies vs. depth (not even in the suppl.). Instead the authors provide all figures vs. the final age model with a rather obscure floating depth scale below.

*We respectfully disagree with the reviewer. In our manuscript, in order to provide a better understanding to the readers, we present the data against the final age model to emphasize the temporal relationships of the proxies and to integrate our findings with broader paleoenvironmental records. This approach allows for a more convincing interpretation of the timing of events and their relevance to the regional and global context.*

*However, the "floating depth scale" in centimeters presented in Fig.3 and Fig.4 as the "depth" has been renamed to "sediment layers". We hope this change clears up any confusion.*

***In summary**, we respectfully disagree with the argument that our study is fundamentally flawed. Combining multiple proxies, we present a coherent story of postglacial environmental change in the western Barents Sea. We have provided interpretations that are well-constrained by data and careful analysis. All presented proxies thus clearly enhance our story regarding meltwater outbursts.*

**Reference**

Bendle, J. and Rosell-Melé, A.: Distributions of UK37 and UK37′ in the surface waters and sediments of the Nordic Seas: Implications for paleoceanography, Geochemistry, Geophysics, Geosystems, 5, Q11013, 10.1029/2004gc000741, 2004.

Berben, S. M. P., Husum, K., Cabedo-Sanz, P., and Belt, S. T.: Holocene sub-centennial evolution of Atlantic water inflow and sea ice distribution in the western Barents Sea, Climate of the Past, 10, 181-198, 10.5194/cp-10-181-2014, 2014.

Blaauw, M. and Christen, J. A.: Flexible paleoclimate age-depth models using an autoregressive gamma process, Bayesian Analysis, 6, 10.1214/11-ba618, 2011.

Groot, D. E., Aagaard-Sørensen, S., and Husum, K.: Reconstruction of Atlantic water variability during the Holocene in the western Barents Sea, Climate of the Past, 10, 51-62, 10.5194/cp-10-51-2014, 2014.

Hughes, A. L. C., Gyllencreutz, R., Lohne, Ø. S., Mangerud, J., and Svendsen, J. I.: The last Eurasian ice sheets – a chronological database and time-slice reconstruction, DATED-1, Boreas, 45, 1-45, 10.1111/bor.12142, 2016.

Lacka, M., Michalska, D., Pawlowska, J., Szymanska, N., Szczucinski, W., Forwick, M., and Zajaczkowski, M.: Multiproxy paleoceanographic study from the western Barents Sea reveals dramatic Younger Dryas onset followed by oscillatory warming trend, Sci Rep, 10, 15667, 10.1038/s41598-020-72747-4, 2020.

Müller, P. J., Kirst, G., Ruhland, G., Von Storch, I., and Rosell-Melé, A.: Calibration of the alkenone paleotemperature index U37K′ based on core-tops from the eastern South Atlantic and the global ocean (60 N-60 S), Geochimica et cosmochimica Acta, 62, 1757-1772, 1998.

Rüther, D. C., Bjarnadóttir, L. R., Junttila, J., Husum, K., Rasmussen, T. L., Lucchi, R. G., and Andreassen, K.: Pattern and timing of the northwestern Barents Sea Ice Sheet deglaciation and indications of episodic Holocene deposition, Boreas, 41, 494-512, 10.1111/j.1502-3885.2011.00244.x, 2012.

Svendsen, J. I., Elverhoi, A., and Mangerud, J.: The retreat of the Barents Sea Ice Sheet on the western Svalbard margin, Boreas, 25, 244-256, DOI 10.1111/j.1502-3885.1996.tb00640.x, 1996.

---

## Author Comment (AC2)

**Reply to the editor's comments**

*All the made responses are in italic blue color*

Based on my own reading of the manuscript, I tend to agree with the assessment of Reviewer 1 and find the author's response to be lacking. While I commend the authors for employing a wide variety of analyses, I am also struggling to see how these data are supporting the interpretation of meltwater pulses. Data resolution in general is problematically low for the majority of the datasets. The methods state the core was sampled at a 1-cm interval, which should result in nearly 150 samples. However the figures appear to show <50 data points for the biomarker data. The methods do not mention the resolution of the analyses. In my opinion, a much higher resolution dataset would be more convincing.

*Thank you for your comment.*

*We understand the problem regarding the data resolution and we acknowledge the concern regarding the resolution of our dataset. However, we'd like to clarify our sampling strategy. Although the core was initially sampled at 1 cm intervals, we applied a standard paleoceanographic approach to select samples for our biomarker analyses. Initially, we analyzed every fourth layer to cover the entire core efficiently. However, recognizing the importance of specific intervals such as meltwater pulses (MP), we increased the resolution by analyzing every second layer within critical sections associated with these events (MP1: 87-97 cm, MP2: 75-81 cm, MP1; 45-55 cm). However, for XRF measurements we analyzed every centimeter of the core. For foraminifera analyses, we followed the same approach as biomarkers, except for the top part of the core (above 20 cm), where sedimentation rates were significantly low, and thus, foraminiferal analysis was conducted for each sediment layer.*

*We will add a detailed description of our sampling strategy and clarify the resolution of the biomarker data in the revised manuscript to address this issue fully.*

I strongly suggest that, at the very least, the manuscript be better organized and with more detailed explanations. I am finding parts that should be in the methods, such as the description of the age-depth modeling routine that was used, in the results. Conversely, the result of the age-depth modeling is contained in the methods. The parameterization of Bacon should also be included in the methods. Figure 3 in the results includes interpreted meltwater events, but these have yet to be introduced in the text. The ordering of information presentation is important. As is, the manuscript is difficult to follow. The term "AW" is never defined in the main text, only in a figure caption after its first usage.

*We apologize for the unclear organization in the manuscript. We will restructure the manuscript to improve the flow of information, making it easier to follow. Specifically:*

*The description of the age-depth modeling routine will be moved from the results section to the methods, along with the detailed parameterization of the Bacon model.*

*Figure 3, which includes the interpreted meltwater events, will be moved to a more appropriate section after the text introduces these events.*

*The term "AW" will be defined at its first mention in the main text as "Atlantic Water," ensuring clarity for readers.*

*These changes will enhance the organization and coherence of the manuscript.*

The methods do not fully explain the methods that were used. For example, section 3.1 (line 110) says the samples were "wet sieved through 100 and 500 μm meshes". This is of course non-standard, and I assume this was not done. Figure 4 shows the >63 μm fraction, so either the statement on line 110 is incorrect, or the corresponding y-axis text label in Figure 4 is incorrect. In Section 3.3 (line 145) dry bulk density is mentioned but there is no description of how it was measured (pycnometer?) In Section 3.4 (line 160), the manuscript states that IRD counts were performed on the >500 μm fraction. IRD counts are typically performed on the >150 μm fraction and using a non-standard size won't allow comparison to other IRD records. Furthermore, the vast about of IRD in icebergs is in the fine fraction, so the statement that melting icebergs "can be ignored" is not really supported (line 311).

*We apologize for the mistake in the text regarding the mesh sizes we used for the sample washing. The samples were freeze-dried and wet-sieved into different size fractions using 63, 100, and 500μm mesh size sieves for the sample washing process. However, the 100 and 500 μm fractions were separated only for the purposes of foraminiferal analyses. For the purposes of sediment grainsize, the entire coarse-grained fraction (>63 μm) was separated. We will clarify this in the revised manuscript.*

*For dry bulk density calculations, we used a common "mass/volume" method ((mass of dry material per cubic centimeter of wet sediment). Specifically, we dried the sediment samples, measured their mass, and divided it by the volume of the sediment slice. We will add this explanation in the revised manuscript.*

*The choice of the fraction for the IRD count strongly depends on the region of the study. We recognize that IRD counts are typically performed on the >150 μm fraction for deep marine sediments because sand content is minimal compared to shallow-depth sediments. However, if the study core site is located in a shallow marine environment and relatively close to the terrestrial environment, larger fractions (>500 μm) are generally used (Rüther et al., 2012; Łącka et al., 2015; Davies et al., 2022; Aagaard-Sørensen et al., 2010). Since our core site is shallow and close to the land (to Bear Island), we opted to use the >500 μm fraction for the IRD count. Such an approach does not allow the direct comparison of IRD fluxes with other records but the general trends in ice rafting can be compared.*

I also believe that the raw data output from the Olympus Vanta M need to be converted to oxides before interpretation, but the methods (Section 3.6) make no mention of this. Was this done? If so, how was it done? The raw output element trends can be very similar without conversion and will vastly change after oxide calculation. Therefore, the similarity between Fe and Ti may be coincidental. The methods must completely and accurately describe all analyses performed in the paper, and what is described gives me serious reservations about the resulting data.

*We used direct element percentages from the Olympus Vanta M instrument and did not convert them to oxides. Instead of using individual element data, we relied on element ratio changes in the sediment profile (e.g., Fe/Ca, Ti/Ca) for interpretation to avoid closed-sum effects (explained in the text; lines 206-207), which can distort the dataset. The striking similarity between Fe and Ti trends throughout the sediment core is not a coincidence but a pattern commonly observed due to the terrestrial matter inflow in various studies across different regions (Devendra et al., 2023; Haug et al., 2001; Caricchi et al., 2018). Similar Fe and Ti trends are associated with terrigenous input, typically reflecting similar depositional processes.*

*The applications of element percentages have already successfully been used for high-resolution time-series studies, stratigraphic correlations, and detailed sedimentary and climatic reconstructions on various time scales (Caricchi et al., 2018; Lucchi et al., 2013; Davies et al., 2022; Bahr et al., 2005; Jaccard et al., 2005).*

All age modeling routines assume that the age determinations reflect the age of sediment deposition. The age reversals, as pointed out by Reviewer 1, do indeed provide evidence of sediment reworking, which in turn does subject the resulting modeled median age to additional uncertainty. I further think that any future version of the manuscript should contain the uninterpreted data plotted versus core depth. I don't see how the figure included in the response to Reviewer 1 supports the statement that "The trends in the multi-proxy records are consistent between the cores." The only two proxies that are the same between the two cores are brassicasterol and SST (why are the other records even shown?), and the trends within the undated sections do not appear similar to me. Brassicasterol absolute values are two orders of magnitude higher in the other core and exhibit a decreasing trend, while there is essentially no trend in the data of the present manuscript. In the SST records, I believe the authors may be referring to the two drops at ~11.5 and ~12.5 ka, but due to the coarseness of the data, it cannot be ruled out that these drops are simply outliers. Hence, my initial comment on increased data resolution.

*Thank you for your comments and suggestions.*

*To increase the age model robustness, we have sent additional samples for dating to the $^{14}C$ laboratory at the Alfred Wegener Institute (AWI), Germany. **We have been assured that additional analyses will be performed within two weeks**. We will present the new dating results in the revised manuscript.*

*Our aim in this manuscript is to show our results/data against the final age model to emphasize the temporal relationships of the proxies and to integrate our findings with broader paleoenvironmental records. However, we will follow your suggestion and include a figure showing the main proxy data plotted against core depth in the revised manuscript as supplementary material, as Reviewer 1 also recommended.*

***Regarding the figure in the response to Reviewer 1**: We used different proxy data from the core JM09-KA11-GC (Berben et al., 2014) showing the comparison with our data for the undated section. We included multiple proxies from JM09-KA11-GC in the figure to demonstrate how the surface water cooling we interpret as meltwater pulses in our undated section is corroborated by nearby records. For example, the decrease in SSTs in our record is consistent with the decline in warm-water planktic species T. quinqueloba and the increase in cold-water planktic species N. pachyderma in the nearby core. Additionally, the stable oxygen isotope data from the nearby core show a decrease in $\delta^{18}O$, further supporting the interpretation of surface freshening during these meltwater pulses.*

*The difference in brassicasterol concentrations could be attributed to the variations in primary productivity or sedimentation rates between the two core sites, affecting the production and accumulation of brassicasterol. We will not compare two brassicasterol records in the revised manuscript.*

**Reference**

Aagaard-Sørensen, S., Husum, K., Hald, M., and Knies, J.: Paleoceanographic development in the SW Barents Sea during the Late Weichselian–Early Holocene transition, Quaternary Science Reviews, 29, 3442-3456, 10.1016/j.quascirev.2010.08.014, 2010.

Bahr, A., Lamy, F., Arz, H., Kuhlmann, H., and Wefer, G.: Late glacial to Holocene climate and sedimentation history in the NW Black Sea, Marine Geology, 214, 309-322, 10.1016/j.margeo.2004.11.013, 2005.

Berben, S. M. P., Husum, K., Cabedo-Sanz, P., and Belt, S. T.: Holocene sub-centennial evolution of Atlantic water inflow and sea ice distribution in the western Barents Sea, Climate of the Past, 10, 181-198, 10.5194/cp-10-181-2014, 2014.

Caricchi, C., Lucchi, R. G., Sagnotti, L., Macrì, P., Morigi, C., Melis, R., Caffau, M., Rebesco, M., and Hanebuth, T. J. J.: Paleomagnetism and rock magnetism from sediments along a continental shelf-to-slope transect in the NW Barents Sea: Implications for geomagnetic and depositional changes during the past 15 thousand years, Global and Planetary Change, 160, 10-27, 10.1016/j.gloplacha.2017.11.007, 2018.

Davies, J., Mathiasen, A. M., Kristiansen, K., Hansen, K. E., Wacker, L., Alstrup, A. K. O., Munk, O. L., Pearce, C., and Seidenkrantz, M.-S.: Linkages between ocean circulation and the Northeast Greenland Ice Stream in the Early Holocene, Quaternary Science Reviews, 286, 10.1016/j.quascirev.2022.107530, 2022.

Devendra, D., Łącka, M., Szymańska, N., Szymczak-Żyła, M., Krajewska, M., Weiner, A. K. M., De Schepper, S., Simon, M. H., and Zajączkowski, M.: The development of ocean currents and the response of the cryosphere on the Southwest Svalbard shelf over the Holocene, Global and Planetary Change, 228, 10.1016/j.gloplacha.2023.104213, 2023.

Haug, G. H., Hughen, K. A., Sigman, D. M., Peterson, L. C., and Röhl, U.: Southward migration of the intertropical convergence zone through the Holocene, Science, 293, 1304-1308, DOI 10.1126/science.1059725, 2001.

Jaccard, S. L., Haug, G. H., Sigman, D. M., Pedersen, T. F., Thierstein, H. R., and Rohl, U.: Glacial/interglacial changes in subarctic north pacific stratification, Science, 308, 1003-1006, 10.1126/science.1108696, 2005.

Łącka, M., Zajączkowski, M., Forwick, M., and Szczuciński, W.: Late Weichselian and Holocene palaeoceanography of Storfjordrenna, southern Svalbard, Climate of the Past, 11, 587-603, 10.5194/cp-11-587-2015, 2015.

Lucchi, R. G., Camerlenghi, A., Rebesco, M., Colmenero-Hidalgo, E., Sierro, F. J., Sagnotti, L., Urgeles, R., Melis, R., Morigi, C., Bárcena, M. A., Giorgetti, G., Villa, G., Persico, D., Flores, J. A., Rigual-Hernández, A. S., Pedrosa, M. T., Macri, P., and Caburlotto, A.: Postglacial sedimentary processes on the Storfjorden and Kveithola trough mouth fans: Significance of extreme glacimarine sedimentation, Global and Planetary Change, 111, 309-326, 10.1016/j.gloplacha.2013.10.008, 2013.

Rüther, D. C., Bjarnadóttir, L. R., Junttila, J., Husum, K., Rasmussen, T. L., Lucchi, R. G., and Andreassen, K.: Pattern and timing of the northwestern Barents Sea Ice Sheet deglaciation and indications of episodic Holocene deposition, Boreas, 41, 494-512, 10.1111/j.1502-3885.2011.00244.x, 2012.

---

## Author Comment (AC3)

Response to RC2

In this manuscript, entitled as "Postglacial environmental changes in the northwestern Barents Sea caused by meltwater outbursts", you tried to detect events of meltwater outburst and/or paleo-tsunami in the northwestern Barents Sea during the last deglaciation period. Your results by using multi-proxies are enough to explain phenomena, but the descriptions for the discussions and conclusion are still unclear.

*Thank you for your feedback. We will work on improving the clarity of the discussions and conclusions in the revised manuscript according to your suggestions below.*

Major correction

Introduction

You mentioned several proxies to clarify your evidences. However, you didn't deeply explain about proxies and examples of their usages. If you can, please add these descriptions to Introduction (or Discussions). Otherwise, readers cannot follow your discussion anymore.

*Thank you for your suggestion. We will add descriptions of the usage of the different proxies used for interpretation in the revised manuscript.*

Organic geochemical analyses.

You used the response factor provided by Belt et al. (2013) for biomarker analysis. However, response factor is different between the machine condition/setting of GC-MS. You used exactly same machine and method to analyze it. If not, it is better to analyze IP25 using GC for making your own response factor to calculate concentrations from GC-MS data.

*In our work, we quantified $IP_{25}$ in the study sediments following the procedure proposed by Belt et al. (2012). However, we did not use the response factors determined by Belt et al. (2012). Instead, $IP_{25}$ concentrations were quantified based on response factors derived from daily GC/MS measurements of relevant standards performed in our laboratory. We will modify the statement in the manuscript (lines 176-177) as follows,*

*"The concentrations of $IP_{25}$ were determined following the procedure described by Belt et al. (2012)."*

There is no introduction for biomarkers such as alkenone (especially C34:4), IP25 and steroids. It seems that you choose more critical indicators among several biomarker proxies. However, the reasons to choose those indicators are still unclear for reader. Especially, you choose C34:4 and PBIP25. Please carefully explain why you choose them for paleoenvironment description.

*We will add an explanation of the biomarkers used, particularly focusing on alkenone ($C_{37:4}$), $IP_{25}$, and steroids in the revised manuscript.*

Reference

Belt, S. T., Brown, T. A., Rodriguez, A. N., Sanz, P. C., Tonkin, A., and Ingle, R.: A reproducible method for the extraction, identification and quantification of the Arctic sea ice proxy IP 25 from marine sediments, Analytical methods, 4, 705-713, 2012.